# Multicast Routing Based on Data Envelopment Analysis and Markovian Decision Processes for Multimodal Transportation

**Mohanad R. Aljanabi** [1,2], **Keivan Borna** [1,*], **Shamsollah Ghanbari** [3] **and Ahmed J. Obaid** [2]

1 Faculty of Mathematical Sciences and Computer, Kharazmi University, Tehran 15719-14911, Iran; mohanad@khu.ac.ir or mohanadr.aljanabi@uokufa.edu.iq
2 Faculty of Computer Science and Mathematics, University of Kufa, Najaf 540011, Iraq; ahmedj.aljanaby@uokufa.edu.iq
3 Department of Computer Science, Faculty of Engineering, Islamic Azad University, Ashtian Branch, Ashtian 142744, Iran; dchpc@ipm.ir
* Correspondence: borna@khu.ac.ir; Tel.: +98-217-763-0040

**Abstract:** In the context of Iraq's evolving transportation landscape and the strategic implications of the Belt and Road Initiative, this study pioneers a comprehensive framework for optimizing multimodal transportation systems. The study implemented a decision-making framework for multimodal transportation, combining data envelopment analysis (DEA) efficiency scores and a Markov decision process (MDP) to optimize transportation strategies. The DEA scores captured decision-making unit (DMU) performance in various aspects, while the MDP rewards facilitated strategic mode selection, promoting efficiency, cost-effectiveness, and environmental considerations. Although our method incurs a total cost approximately 29% higher than MRMQoS, it delivers a nearly 26% reduction in delay compared to MCSTM. Despite MRMQoS yielding an 8.3% higher profit than our approach, our proposed scheme exhibits an 11.7% higher profit compared to MCSTM. In terms of computational time, our method achieves an average CPU time positioned between MCSTM and MRMQoS, with MCSTM showing about 1.6% better CPU time than our approach, while our method displays a 9.5% improvement in computational time compared to MRMQoS. Additionally, concerning $CO_2$ emissions, the proposed model consistently outperforms other models across various network sizes. The percentage decrease in $CO_2$ emissions achieved by the proposed model is 7.26% and 31.25% when compared against MRMQoS and MCSTM for a network size of 25, respectively.

**Keywords:** multicast routing; multimodal transportation; data envelopment analysis; decision-making unit; markovian decision process; Iraq transportation network

## 1. Introduction

Multimodal transportation optimization has emerged as a crucial strategy to address the complex challenges faced by nations worldwide, and Iraq is no exception. In the context of Iraq's transportation landscape, multimodal transportation optimization stands as a beacon of hope, promising to alleviate unique challenges while leveraging the nation's geographical advantage for economic growth and development. Strategically positioned in the heart of the Middle East, Iraq holds immense potential for a robust multimodal transportation system, facilitated by vital waterways like the Persian Gulf and an extensive road network linking major cities and neighboring nations. This geographic advantage, when properly harnessed, can elevate Iraq's transportation network to new heights [1–3].

Multimodal transportation integrates various modes such as road, rail, air, and sea to efficiently move goods and people. Iraq's diverse geography, including mountains, deserts, and a substantial coastline along the Persian Gulf, necessitates a strategic approach to transportation. Recognizing the varying needs of different regions, Iraq embraces multimodal transportation as a flexible solution [4–6]. This integrated approach proves vital for

economic development and post-conflict reconstruction, ensuring efficient transportation networks for rebuilding infrastructure.

The economic backbone of Iraq, particularly its reliance on oil exports, underscores the importance of efficient transportation links from production centers to export terminals. Multimodal transportation facilitates seamless movement, contributing to overall economic resilience and growth. Additionally, Iraq's geographical location, surrounded by neighboring countries, emphasizes the significance of regional connectivity. Adopting multimodal transportation enhances collaboration, cross-border trade, and economic ties, fostering a more interconnected and prosperous region. Efficiency, cost reduction, and security are additional advantages offered by multimodal transportation, especially pertinent in mitigating historical security challenges by diversifying transportation routes [4–6].

From an environmental perspective, multimodal transportation contributes to sustainability efforts by reducing overall greenhouse gas emissions. By strategically utilizing different modes based on their efficiencies and environmental impacts, Iraq can make strides toward a more sustainable and eco-friendly transportation system. Addressing urban congestion, particularly in major cities like Baghdad, is another benefit. Integrating public transit alongside other modes can alleviate traffic congestion, enhancing the quality of life for urban residents. Moreover, well-developed transportation infrastructure, including air travel and roads, promotes tourism, leading to economic growth, revenue generation, and increased employment opportunities. In times of humanitarian crises, efficient multimodal transportation is indispensable for delivering timely and essential humanitarian aid to affected populations [4–6]. This comprehensive approach ensures that aid reaches its destination promptly, contributing to the well-being of those in need. In essence, Iraq's adoption of multimodal transportation reflects a holistic strategy that addresses diverse needs, promotes economic development, enhances security, and fosters sustainability across various facets of the nation's progress.

In light of these challenges, the integration of various transportation modes through approaches like data envelopment analysis (DEA) and hidden Markov models (HMM) becomes instrumental [7–11]. These methods offer tailored solutions that can address Iraq's transportation woes and unlock the full potential of its transportation network. In a context where infrastructure quality profoundly influences safety and efficiency, DEA employed for efficiency evaluation utilizes real-time data to assess the efficiency of each transportation arc. Subsequently, targeted measures can be recommended to enhance efficiency and safety, addressing critical aspects in a country where the quality of infrastructure significantly impacts transportation safety and efficiency [7–11]. In tandem, the application of HMM for predictive modeling proves instrumental in transforming transportation dynamics. HMM predicts transportation costs and $CO_2$ emissions based on historical data, providing valuable insights for decision-making. Utilizing historical data for training, the HMM enhances prediction accuracy and aids in optimizing transportation planning [7–11].

This paper marks a paradigm shift in multimodal transportation decision-making by introducing a dynamic fusion of DEA and HMM. The study pioneers a holistic framework that simultaneously evaluates financial and environmental impacts, showcasing a novel approach in the cargo sector. The real-time adaptation to efficiency scores and predictive modeling further contributes to the unprecedented depth and breadth of this research. The primary contribution lies in emphasizing the crucial need for adopting multimodal transportation optimization in Iraq, utilizing advanced techniques, specifically DEA and HMM, as invaluable assets for addressing the complex challenges within the nation's transportation sector. These tailored approaches are recognized for their capacity to enhance infrastructure, increase transportation efficiency, reduce emissions, and ensure the safety and security of transportation routes.

Moreover, the paper advocates for the integration of these advanced techniques as pivotal elements in Iraq's journey toward establishing a modern, efficient, and sustainable transportation network. It positions DEA and HMM as instrumental tools in unlocking Iraq's full economic potential through improved logistics and transportation systems.

The acknowledgment of intricate environmental impacts associated with transportation and logistics systems is a notable aspect, with an emphasis on proactive environmental management strategies. The paper urges a transition from mere regulatory compliance to a forward-thinking approach, identifying negative interactions, impact types, and alternative methods to control pollution and natural resource degradation, aligning with contemporary environmental standards and sustainable practices [3,4,12–14].

The structure of this paper is as follows: Section 2 presents the literature review, offering a comprehensive overview of existing studies and findings related to multimodal transportation networks. Section 3 explains the system model and design methodology. Section 4 presents a performance evaluation of the paper, including simulation environmental and simulation results evaluating the performance of the multimodal transportation network system using various metrics. Finally, Section 5 concludes the paper by summarizing the key findings and contributions of the study.

## 2. Literature Review

In the contemporary context, the transport industry emerges as a significant contributor to greenhouse gas emissions, responsible for roughly one-third of global emissions. As the freight transportation sector experiences continuous growth, there is an urgent requirement to create transportation networks that are both sustainable and effective. Multimodal transportation networks offer a promising solution by seamlessly integrating various modes, including rail, road, and water, aiming to reduce costs and improve supply chain performance. Nevertheless, the intricate task of designing such networks is beset with challenges, necessitating a comprehensive approach. Past research has employed various methodologies, such as linear programming, integer programming, and metaheuristic algorithms, to achieve a balance between multiple objectives in optimizing transportation [15–17].

Multicasting, a crucial element in computer networks, involves the simultaneous transmission of identical data from a source to a group of destinations. Multicast routing, vital for multimedia information transmission, involves selecting a routing tree from a source to encompass all destinations, addressing the NP-complete Steiner tree problem. The challenge lies in constructing an optimal multicast tree with minimal cost, considering quality of service (QoS) requirements like delay, delay jitter, bandwidth, and packet loss inherent in multimedia communications. To address these constraints, multicast routing introduces the concept of a constrained Steiner tree, an NP-complete problem tackled by various heuristic algorithms [18–20].

Taboada et al. [8] evaluate the efficiency and sustainability of urban rail transit (URT) through a two-stage methodology involving exploratory data analytics (EDA) and data envelopment analysis (DEA). EDA characterizes URT efficiency and sustainability using existing and suggested indicators, while DEA assesses URT efficiency through two original models. The proposed methodology undergoes experimental validation using open data from the Transport for London (TfL) URT network. Antunes et al. [9] address deficiencies in previous road transportation sustainability research by considering epistemic uncertainty related to innovation and research and development (R&D) expenditure impact on pollutant emissions performance. They introduce a TEA-IS model for assessing road transportation sustainability performance in 29 Chinese provinces over a 14-year period, employing a hybrid DEA-TOPSIS approach and machine learning techniques. Results reveal high synergy in Chinese provinces, suggesting the need for favorable policies to enhance innovation and attract foreign direct investment.

Zhang et al. [10] highlight the transportation network design problem (TNDP), involving strategic criteria selection to enhance an existing transportation network amid increasing traffic demand. The authors employ a hidden Markov model and equilibrium optimizer (EO) for resolution, demonstrating the novel method's superior effectiveness through comparative analysis on a test network. Fotuhi et al. [21] present a robust mixed-integer linear program (MILP) supporting railway operators in making informed decisions

about the expansion of their intermodal networks. The model addresses uncertainties related to demand and supply, incorporating budgetary constraints and capacity limitations. A hybrid GA, utilizing column generation and multimodal shortest path label-setting algorithms, tackles the complex MILP. Raayatpanah [11] proposes an innovative approach to construct multicast trees incorporating multiple QoS parameters, utilizing data envelopment analysis techniques. The effectiveness of the proposed method is demonstrated through numerical examples.

Unmanned aerial vehicles (UAVs) equipped with high-definition cameras play a crucial role in efficiently collecting comprehensive road data from diverse angles. However, their limited energy capacity poses challenges for sustained operation in such tasks. Therefore, optimizing UAV path planning to minimize energy consumption is paramount. To address this challenge, Kong et al. [22] proposed a novel approach called the multi-agent deep deterministic policy gradient-based (MADDPG) algorithm for UAV path planning (MAUP). Our method focuses on optimizing energy consumption and memory usage through targeted optimizations.

In previous studies, the effectiveness of optimizing multimodal transportation networks has been highlighted by utilizing hidden Markov models (HMM) and data envelopment analysis (DEA) [7–11]. These approaches have proven successful in reducing transportation costs and improving service levels. Researchers have explored challenges and opportunities in multimodal freight transportation, with a specific focus on enhancing efficiency, cost-effectiveness, and environmental sustainability. Mathematical models and algorithms, related to DEA and HMM, have been developed to enhance overall performance, sustainability, and logistics in the field [7–11]. Nevertheless, establishing an efficient multicast tree requires consideration of various optimization objectives, including maximizing network throughput, minimizing latency, delay, jitter, cost, power consumption, and error rates. This paper delves into the intricate domain of multicast routing within multimodal transportation networks. It extensively explores the utilization of DEA for evaluating the efficiency of network arcs. Furthermore, it integrates HMM to predict transportation costs and $CO_2$ emissions accurately. The core contribution lies in the construction of a multicast tree, aligning with QoS constraints, achieved through formulating the problem as a MILP problem. Additionally, the paper proposes an innovative approach grounded in Markovian decision processes to optimize policies tailored for the complex landscape of multimodal transportation.

## 3. System Methodology

The system methodology includes the multicast routing for multimodal transportation, DEA for arc efficiency evaluation, HMM for predicting transportation cost and $CO_2$ emissions, construction of multicast tree with QoS constraints by formulating the problem as an integer linear programming (ILP) problem, and Markovian decision processes-based optimized policy for multimodal transportation.

### 3.1. Multicast Routing and QoS Constraints

Multicast routing involves the concurrent transmission of information from a single source to multiple destinations in a network. In the context of multimodal transportation, this concept extends beyond data to encompass the efficient routing of goods via various transportation modes. The significance of multicast routing lies in its ability to streamline the delivery process, reducing costs and improving overall system efficiency. In this section, we delve into the key concepts of multicast routing, explore the challenges posed by the Steiner tree problem, and emphasize the importance of addressing quality of service (QoS) constraints. Mathematically, let $G = (V, E)$ represent the transportation network graph, where V is the set of cities and E is the set of arcs connecting these cities, as shown in Figure 1. The multicast routing problem can be formalized as finding a tree that spans a subset of V and minimizes the overall cost or distance. The business has access to four modes of transportation in Iraq: truck, rail, ship, and air, which can be utilized for

product transportation. Each arc $(u, v) \in E$ in the transportation network may have multiple transportation modes available for use, and for each mode $m \in M$ on arc $(u, v)$, a number of possible departure times are listed by $(u, v, m)$ as shown in Figure 1. The transportation capacity of each arc is mode-dependent, and upload and download times at switch points are taken into account to estimate transportation time. It is assumed that no return transportation is considered in this study [18,23].

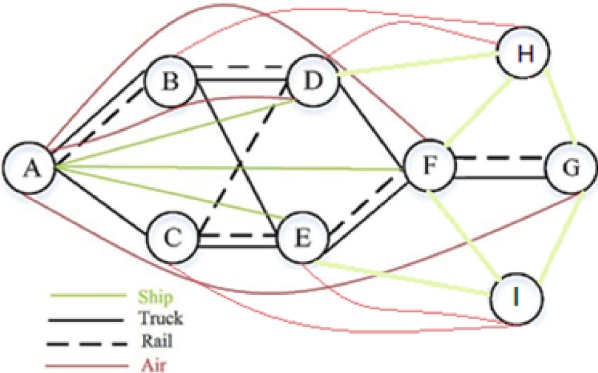

**Figure 1.** Multimodal Transportation Network in Iraq. Nodes A–I are interconnected using sea, raod, railways and air transportation sources.

The received customer orders are represented by $|K|$, with varying demand due dates, quantities, and priority levels. While maintaining the service quality for urgent orders and VIP customers, the corporation may be willing to delay the delivery of some non-urgent orders during periods of high demand. The goal is to reduce the overall transportation, carbon emissions, and delay costs subject to the supply and demand constraints. Historical data collected from the Ministry of Transportation, Iraq are used to calculate the transportation costs and $CO_2$ emissions for each arc and mode combination.

The Steiner tree problem is a fundamental challenge in multicast routing. It involves finding the minimal tree that connects a subset of vertices, known as terminal nodes, in a network. In the context of multimodal transportation, the Steiner tree problem corresponds to determining the optimal routes that connect various cities while considering QoS parameters. Introducing QoS constraints adds a layer of complexity to the Steiner tree problem. QoS parameters may include constraints on delay, bandwidth, packet loss, and other factors crucial for efficient transportation. Mathematically, the constrained Steiner tree problem can be expressed as follows:

$$Min \sum_{(i,j) \in A} c_{ij} \cdot x_{ij} \tag{1}$$

Subject to constraints such as $\sum_j x_{ij} = 1$ and $\sum_{(i,j) \in A} q_{ij} \cdot x_{ij} \leq Q$, where $c_{ij}$ represents the cost of traveling from city $i$ to city $j$, $q_{ij}$ denotes the QoS parameter associated with the arc $(i, j)$, and $Q$ is the specified limit for the QoS parameter. Traditional multicast algorithms often fall short in meeting the demands of modern multimodal transportation networks. These algorithms may neglect QoS constraints or fail to consider the dynamic nature of transportation systems. As a result, there is a pressing need for a more sophisticated approach that integrates advanced techniques such as DEA and HMM to enhance the accuracy and efficiency of multicast routing. In the following sections, we will explore how DEA and HMM contribute to addressing these limitations, providing a robust framework for optimizing the multimodal transportation network under various constraints and uncertainties.

### 3.2. Efficiency Assessment of Transportation Arcs Using DEA

DEA emerges as a powerful non-parametric methodology for the comprehensive evaluation of DMUs in the intricate realm of multimodal transportation networks. Originating

from the seminal work by Charnes et al. [24], DEA provides an effective means to gauge the relative efficiency of peer units without requiring a predetermined functional form for a production function. The production frontier, constructed as a convex and piecewise function, is formulated by the linear combination of efficient units. Consider a scenario with p DMUs to be evaluated, indexed by $j \in \{1, \ldots, p\}$. Each DMU is presumed to consume $k$ input levels to produce s different outputs. The input and output vectors for DMUj are denoted as $x_j = \left( x_{1j}, \ldots, x_{kj} \right)$ and $y_j = (y_{1j}, \ldots, y_{sj})$, respectively. All components of vectors $x_j$ and $y_j$ are non-negative for all DMUs, with each DMU having at least one strictly positive input and output. The relative efficiency score of DMU$_o$ is expressed as:

$$\theta_o = max_{u,v} \left( \frac{\sum_{r=1}^{s} u_r y_{ro}}{\sum_{i=1}^{k} v_i x_{io}} \right) \tag{2}$$

where $u_r$ and $v_i$ are the non-negative weights associated with output $r$ and input $i$, respectively. It is essential to ensure that these weights, when applied to other DMUs, do not result in an efficiency value exceeding one [25]. This requirement can be formally stated through the following constraints:

$$\left( \frac{\sum_{r=1}^{s} u_r y_{rj}}{\sum_{i=1}^{k} v_i x_{ij}} \right) \leq 1, \ j = 1, 2, \ldots p \tag{3}$$

Therefore, the relative efficiency score of DMUo is obtained by the following linear fractional model,

$$\theta_o^* = Max \left( \frac{\sum_{r=1}^{s} u_r y_{ro}}{\sum_{i=1}^{k} v_i x_{io}} \right) \tag{4}$$

subject to the constraints,

$$\left( \frac{\sum_{r=1}^{s} u_r y_{rj}}{\sum_{i=1}^{k} v_i x_{ij}} \right) \leq 1, \ j = 1, 2, \ldots p \tag{5}$$

$$v_i \geq \epsilon \ i = 1, 2, \ldots k \tag{6}$$

$$u_r \geq \epsilon \ r = 1, 2, \ldots s \tag{7}$$

In this context, the parameter $\epsilon > 0$ serves as a non-Archimedean element ensuring the existence of strongly efficient solutions [25]. It is important to highlight that $0 < \theta_o^* \leq 1$, and a DMU becomes efficient when $\theta_o^* = 1$. Each DMU is assessed based on its optimal weight. The outcomes of DEA models involve identifying the hyperplanes that delineate an envelope surface or Pareto frontier. It is noteworthy that an efficient DMU will be positioned on the Pareto frontier. The initial fractional model can be converted into the subsequent linear program using the Charnes–Cooper transformation [26].

$$\theta_o^* = Max \sum_{r=1}^{s} u_r y_{ro} \tag{8}$$

subject to the constraints,

$$\sum_{i=1}^{k} v_i x_{io} = 1 \tag{9}$$

$$\sum_{r=1}^{s} u_r y_{rj} - \sum_{i=1}^{k} v_i x_{ij} \leq 0, \ j = 1, 2, \ldots p \tag{10}$$

$$v_i \geq \epsilon \ i = 1, 2, \ldots k \tag{11}$$

$$u_r \geq \epsilon \ r = 1, 2, \ldots s \tag{12}$$

Equation (8) represents the primary CCR model, introduced by Charnes, Cooper, and Rhodes. Let $u_r^*(r = 1, 2, \ldots s)$ and $v_i^*(i = 1, 2, \ldots k)$ represent the optimal output and input weights in the aforementioned model. The term $\theta_o^* = \sum_{r=1}^{s} u_r^* y_{ro}$ quantifies the

best relative efficiency score of $DMU_o$, referred to as the CCR-efficiency of $DMU_o$. A $DMU_o$ is considered CCR-efficient if $\theta_o^* = 1$; otherwise, it is classified as CCR-inefficient. $\theta_{jo} = \left( \frac{\sum_{r=1}^{s} u_r^* y_{rj}}{\sum_{i=1}^{k} v_i^* x_{ij}} \right)$ is referred to as the cross-efficiency value of $DMU_j$, which reflects the peer evaluation of $DMU_o$ to $DMU_j$ ($j = 1, 2, \ldots p; j \neq o$). As a result, we generate a *pxp* matrix where the diagonal elements represent the CCR-efficiency scores of DMUs, and the remaining entries provide the cross-efficiency scores. To facilitate cross-evaluation, the average cross-efficiency score is computed in each column, offering a distinctive ranking of DMUs and mitigating impractical weight configurations [27]. However, adjustments to the matrix of cross efficiency may be necessary to address (2) due to the presence of alternative input and output weights. The non-uniqueness of optimal weights poses a challenge to cross-efficiency evaluation. To address this, Sexton et al. [28] and Doyle and Green introduced a secondary objective. Doyle and Green [29] introduced aggressive and benevolent models, with the aggressive model formulated as follows:

$$\left( \sum_{j=1, j\neq o}^{p} y_{rj} \right) \tag{13}$$

subject to the constraints,

$$\sum_{i=1}^{k} v_i \left( \sum_{j=1, j\neq o}^{p} x_{ij} \right) = 1, \tag{14}$$

$$\sum_{r=1}^{s} u_r y_{ro} - \theta_o^* \sum_{i=1}^{k} v_i x_{io} = 0, \tag{15}$$

$$\sum_{r=1}^{s} u_r y_{rj} - \sum_{i=1}^{k} v_i x_{ij} \leq 0, j = 1, 2, \ldots p; \ j \neq o \tag{16}$$

$$v_i \geq \epsilon \ \ i = 1, 2, \ldots k \tag{17}$$

$$u_r \geq \epsilon \ \ r = 1, 2, \ldots s \tag{18}$$

where $\theta_o^*$ is the CCR-efficiency of DMUo derived from (8).

### 3.3. HMM for Predicting Transportation Costs and $CO_2$ Emissions

HMMs are probabilistic models widely used in various fields, including predictive modeling. HMMs are particularly effective in scenarios where underlying processes involve hidden states that influence observed outcomes. In the context of multimodal transportation, HMMs can be employed to model the dynamic nature of transportation costs and $CO_2$ emissions. An HMM consists of a set of hidden states, observable symbols emitted from these states, and transition probabilities between states. In the transportation domain, hidden states may represent different operating conditions, and observable symbols could correspond to cost and emission levels. Let us define the components of an HMM for predicting transportation costs (TC) and $CO_2$ emissions (CE).

- Hidden States: $S = \{S_1, S_2, \ldots, S_N\}$, representing different operating conditions (e.g., normal operation, peak demand, maintenance).
- Observable Symbols: $O = \{o_1, o_2, \ldots, o_M\}$, indicating levels of transportation costs and $CO_2$ emissions.
- State Transition Probabilities: $a_{ij} = P(q_t = S_j \mid q_{t-1} = S_i)$, where $q_t$ represents the hidden state at time $t$.
- Emission Probabilities: $b_j(k) = P(o_k att \mid q_t = S_j)$, where $b_j(k)$ is the probability of observing symbol $o_k$ when in state $S_j$.
- Initial State Probabilities: $\pi_i = P(q_1 = S_i)$, representing the probability of starting in state $S_i$.

Given a sequence of observed symbols $= o_1, o_2, \ldots, o_T$, the goal is to find the most likely sequence of hidden states $S = S_1, S_2, \ldots, S_T$ that generated the observed sequence. This is achieved through the Viterbi algorithm,

$$\delta_t(j) = max_i \left[ \delta_{t-1}(i) \cdot a_{ij} \right] \cdot b_j(o_t) \tag{19}$$

$$\psi_t(j) = argmax_i \left[ \delta_{t-1}(i) \cdot a_{ij} \right] \tag{20}$$

The most likely state sequence is then obtained by backtracking through the computed values. Training the HMM involves estimating the model parameters ($a_{ij}$, $b_j(k)$, $\pi_i$) based on historical data. The expectation–maximization (EM) algorithm is commonly employed for this purpose.

Let $Q = \{q_1, q_2, \ldots, q_T\}$ represent the hidden state sequence corresponding to observed symbols $O$. The EM algorithm iteratively maximizes the expected log-likelihood:

$$Q(\lambda \mid O) = \sum_Q P(Q \mid O, \lambda) \cdot log P(O, Q \mid \lambda) \tag{21}$$

where $\lambda$ represents the set of model parameters.

The update equations for the HMM parameters are as follows:

- Transition Probabilities: $a_{ij} = \frac{\sum_{t=2}^{T} \xi_t(i,j)}{\sum_{t=1}^{T-1} \gamma_t(i)}$;
- Emission Probabilities: $b_j(k) = \frac{\sum_{t=1}^{T} \gamma_t(j) \cdot \delta(o_t - k)}{\sum_{t=1}^{T} \gamma_t(j)}$;
- Initial State Probabilities: $\pi_i = \gamma_1(i)$.

Here, $\xi_t(i,j)$ is the probability of transitioning from state $S_i$ to $S_j$ at time $t$, $\gamma_t(i)$ is the probability of being in state $S_i$ at time $t$, and $\delta(\cdot)$ is the Kronecker delta function.

By utilizing historical data, the HMM is trained to capture the underlying patterns in transportation costs and $CO_2$ emissions, enabling accurate predictions in real-time scenarios. This approach enhances the system's ability to anticipate changes, optimize resource allocation, and improve overall decision-making in the multimodal transportation network.

*3.4. Construction of Multicast Tree with QoS Constraints*

The construction of a multicast tree with quality of service (QoS) constraints is crucial for optimizing the multimodal transportation network. To achieve this, we formulate the problem as an integer linear programming (ILP) problem. The decision variables and parameters related to construction of multicast tree are as follows.

$x_{ij}$: Binary variable indicating whether arc $(i, j)$ is included in the multicast tree (1 if included, 0 otherwise).
$y_i$: Binary variable indicating whether city $i$ is selected as part of the multicast tree (1 if selected, 0 otherwise).
$c_{ij}$: Transportation cost associated with arc $(i, j)$.
$ce_{ij}$: Predicted $CO_2$ emissions associated with arc $(i, j)$.
$q_{ij}$: QoS parameter associated with arc $(i, j)$.
$C_i$: Capacity constraint for city $i$.
$D_j$: Demand constraint for destination city $j$.
$R_T$: Risk threshold for $CO_2$ emissions.
$DC_{ij}$ represents the delay cost associated with arc $(i, j)$.
$d_{ij}$ is a binary variable indicating whether a delay occurs on arc $(i, j)$.

The overall objective is to minimize the total transportation cost, $CO_2$ emissions, and delay cost associated with arc $(i, j)$ while meeting QoS, supply, and demand constraints. The ILP formulation is as follows:

$$Min \sum_{(i,j) \in A} c_{ij} \cdot x_{ij} + Min \sum_{(i,j) \in A} ce_{ij} \cdot x_{ij} + Min \sum_{(i,j) \in A} DC_{ij} \cdot d_{ij} \tag{22}$$

*3.5. Integration of DEA Efficiency Scores and HMM Predictions into ILP*

The ILP formulation incorporates the efficiency scores obtained from DEA and the predictions from the HMM. This integration allows for a comprehensive optimization approach that considers both historical efficiency and dynamic predictions.

$$Maximize \ \rho_{ij} \tag{23}$$

subject to $\frac{\sum_{k=1}^{n} \lambda_k \cdot x_{kj}}{\sum_{k=1}^{m} u_k \cdot ce_{kj}} \leq \rho_{ij}$.

This constraint ensures that the inclusion of an arc in the multicast tree is influenced by its historical efficiency ($\rho_{ij}$) obtained from DEA.

$$\textit{Maximize } \delta_{ij} \tag{24}$$

subject to $\delta_{ij} \leq q_{ij}$.

Here, $\delta_{ij}$ represents the QoS parameter associated with the arc $(i, j)$, and the constraint ensures that the QoS parameter is satisfied.

*3.6. Constraints in the ILP formulation*

1.  Supply and Demand Constraints

$$\sum_j T_{ij}^{mn} \leq C_i^m \tag{25}$$

$$\sum_j T_{ij}^{mn} \geq D_j^n \tag{26}$$

These constraints ensure that the total amount of goods transported from each origin city does not exceed the capacity of the transportation mode and meets the demand for each shipment.

2.  Non-negativity Constraints

$$T_{ij}^{mn}, X_{ij}^{mn}, Y_j, d_{ij} \geq 0 \tag{27}$$

These constraints ensure that decision variables representing quantities, selections, and delays are non-negative.

3.  Mode Selection Constraints

$$\sum_m X_{ij}^{mn} = 1 \tag{28}$$

This constraint ensures that for each shipment from origin $i$ to destination $j$, only one transportation mode is selected.

4.  DEA Efficiency Constraints

$$\rho_{ij} = \frac{\sum_{k=1}^{n} \lambda_k \cdot x_{kj}}{\sum_{k=1}^{m} u_k \cdot ce_{kj}} \tag{29}$$

This constraint ensures that the DEA efficiency scores are calculated based on the historical efficiency and predicted $CO_2$ emissions.

5.  QoS Constraints for Multicast Routing

$$\delta_{ij} \leq q_{ij} \cdot x_{ij} \tag{30}$$

These constraints ensure that the QoS parameters for each arc in multicast routing meet specified parameters.

The incorporation of these constraints reflects a contemporary optimization model that considers historical efficiency, predictive accuracy, and QoS parameters, enabling a robust and adaptable decision-making framework. The inclusion of Markovian decision-making using DEA and HMM enhances the model's ability to capture dynamic changes in the transportation network, ensuring optimal and reliable solutions in the context of the Iraq Belt and Road Initiative project.

6.  Risk Threshold Constraint

$$\sum_{ij} \delta_{ij} x_{ij} \leq R_T \tag{31}$$

This constraint controls the ratio of the product of the QoS parameter and risk value to the demand for final goods.

*3.7. Markovian Decision Processes-Based Optimized Policy for Multimodal Transportation*

In the realm of multimodal transportation, the decision-making process involves two crucial aspects: evaluating the efficiency of transportation modes and devising an optimal policy for mode selection. This mathematical model seamlessly integrates data envelopment analysis (DEA) for efficiency assessment and a Markov decision process (MDP) for policy optimization. The decision variables used in this process are $\theta_r$, which is efficiency score for mode $r$, and weight assigned to each mode $k$ denoted by $\lambda_k$. The objective is to maximize $\theta_r$ with respect to following constraints.

Ensure that the weighted sum of infrastructure costs for all modes is less than or equal to $\theta_r$ times the infrastructure cost of mode,

$$\sum_k \lambda_k IC_k \leq \theta_r IC_r \tag{32}$$

Ensure that the weighted sum of passenger and freight capacities for all modes is greater than or equal to the respective capacities of mode,

$$\sum_k \lambda_k PC_k \geq PC_r \tag{33}$$

$$\sum_k \lambda_k FC_k \geq FC_r \tag{34}$$

$\lambda_k$ values are non-negative: $\lambda_k \geq 0$ for all k.

The MDP is employed to formulate an optimal policy for mode selection over time, considering rewards associated with efficiency, cost, and emissions.

$$Maximize \sum_{t=0}^{\infty} \gamma^t \cdot R(s_t, \pi(s_t)) \tag{35}$$

where $R(s, a)$ is the reward function for state $s$ and action $a$, $P(s' \mid s, a)$ is the transition probability from state $s$ to state $s'$ given action $a$, $V(s)$ is the value function for state $s$, and $\pi(s)$ is the policy function indicating the recommended action for state $s$. The rewards are computed based on DEA efficiency scores and associated costs and emissions. The Bellman equation, which can also be considered as policy improvement, is mentioned in (35) and (36).

$$V(S) = max_{a \in A} \left( R(s, a) + \gamma \sum_{s' \in S} P(s' \mid S, a) \cdot V(s') \right) \tag{36}$$

$$\pi(S) = arg\ max_{a \in A} \left( R(s, a) + \gamma \sum_{s' \in S} P(s' \mid S, a) \cdot V(s') \right) \tag{37}$$

We repeatedly evaluate and refine the policy until convergence. This model essentially balances the short-term efficiency of transportation modes through DEA with the long-term benefits derived from the MDP. It offers a comprehensive approach to decision-making in multimodal transportation, aligning efficiency considerations with strategic, forward-looking policy choices.

## 4. Performance Evaluation of Proposed Scheme

*4.1. Simulation Environment*

The dataset employed in this study is sourced from the Ministry of Transportation, Iraq which is the General Company for (land, train, water, and air transport) [30]. The temporal scope of the data spans from 2017 to 2022, offering a comprehensive insight into the dynamics of the transportation sector over a six-year period. For the General Company for Navy Transport, detailed emissions data were collected, encompassing two distinct fuel types: 'Diesel(oil)' and 'Gasoline'. This information includes the quantity of fuel consumed

per liter, carbon dioxide emissions per ton ($CO_2$), hydrocarbon emissions per ton (HC), and nitrogen oxide emissions per ton ($NO_2$). The dataset reveals a nuanced pattern of emissions, shedding light on the environmental impact of the company's operations, particularly in terms of nitrogen oxides (Nox), sulfur dioxide ($SO_2$), and methane ($CH_4$). Furthermore, the dataset encompasses the operational and environmental metrics of Iraqi Airways, covering the years 2017 to 2021. Key parameters include the number of trips, and gas emissions measured in international tons. The dataset indicates a significant growth in the number of trips over the years, potentially indicative of the airline's expansion. Simultaneously, the gas emissions data suggest a considerable environmental impact associated with the airline's operations. Additionally, the dataset extends to the transportation sector beyond air travel, incorporating data from trucks, railways, and airplanes. For trucks, the dataset includes monthly quantities transported, distances covered, and associated costs. Railway data involve monthly quantities transported, revenues, ton-kilometers, and total costs. Similarly, airplane data incorporate loaded and unloaded weights, loaded and unloaded costs, and total costs.

These data serve as the foundation for a sophisticated approach integrating DEA and HMM in the context of multimodal transportation within the supply chain industry. To tackle the intricacies of multimodal transportation optimization, the dataset undergoes strategic segmentation into three distinct groups. This segmentation facilitates an iterative refinement of our algorithm, a critical step in achieving an optimal solution. The incorporation of DEA, a non-parametric method, allows for the assessment of relative efficiencies among DMUs consuming multiple inputs to produce multiple outputs. This evaluation is pivotal for identifying optimal transportation modes within the supply chain. Concurrently, HMM, a probabilistic model, captures the dynamic nature of the transportation network. By modeling the underlying hidden states and observable emissions, HMM provides valuable insights into the evolving conditions of the transportation system. The integration of DEA and HMM enhances the model's capabilities, allowing stakeholders to make informed decisions based on a holistic evaluation of efficiency, cost-effectiveness, and the dynamic nature of the transportation network. We have given some of the simulation parameters which are specifically used in different transportation modes of supply chain networks in Iraq in Table 1.

**Table 1.** Simulation parameters of different transportation modes [2,12–14].

| Parameters | Transportation Mode and Values | | | |
| --- | --- | --- | --- | --- |
| | **Trucks** | **Rail** | **Ship** | **Air** |
| Activation cost of supplier per vehicle (IQD) | [26,700–32,040] | [30,260–32,040] | [26,700–30,260] | [20,000–26,700] |
| Variable transportation cost of shipment per km per vehicle (IQD) | [13.35–15.13] | [6.23–13.35] | [2.67–8.9] | [0.445–1.335] |
| Carbon emissions per kilometer per weight for transportation mode m | [0.15–0.18] | [0.07–0.17] | [0.02–0.04] | [0.005–0.02] |
| Total volume capacity of a vehicle/container in m$^3$ | [40–70] | [40–70] | [45–60] | [15–40] |
| Total weight capacity of a vehicle/container in kg | [18,000–20,000] | [18,000–20,000] | [20,000–22,000] | [18,000–20,000] |
| Maximum iterations | | 100 | | |
| Average customer demand | | [400–600] | | |
| Customer demand variance | | [100–200] | | |
| Supplier capacity for the final product | | 4~6 | | |
| The production cost of the final product | | [0.5–5] | | |
| Carbon emissions for the final product | | [0.02–0.08] | | |
| Supplier risk | | [0.05–0.20] | | |
| Supplier/plant/customer locations | | Randomly distributed | | |
| Weight of the final product | | [10–25] | | |

### 4.2. Simulation Results

We implemented a decision-making framework for multimodal transportation systems by combining DEA and an MDP. The process begins by computing DEA efficiency scores

for each DMU in the multimodal transportation network. The DEA efficiency scores capture the performance of each DMU in terms of infrastructure cost, fuel consumption, number of employee, passenger capacity, freight capacity, and $CO_2$ emissions. Subsequently, we formulated an MDP to optimize decision-making in the transportation network. The MDP considers three actions, each corresponding to a specific strategy or transportation mode. The rewards for these actions are influenced by the DEA efficiency scores, infrastructure costs, and $CO_2$ emissions. The objective is to find an optimal policy that balances efficiency with the associated costs and emissions. The MDP is solved iteratively using policy iteration, where the policy is evaluated and improved until convergence. Figure 2 depicts the DEA efficiency score for 13 transportation modes, i.e., A: Truck, B: Rail, C: Ship, D: Air, E: Truck + Rail, F: Truck + Ship, G: Truck + Air, H: Rail + Ship, I: Rail + Air, J: Ship + Air, K: Truck + Rail + Ship, L: Truck + Rail + Air, M: Truck + Rail + Ship + Air. It is quite evident from Figure 2 that transportation modes C, F, J, and K outperform all other modes keeping in view the demographical landscape of Iraq. Depending on the type of shipment/cargo, the combination of truck, rail, and sea is more suitable for Iraq, as indicated by DEA efficiency score.

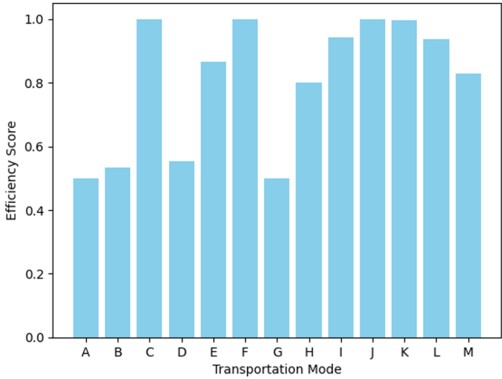

**Figure 2.** DEA efficiency score of different transportation modes.

In Figure 3, the MDP rewards for 13 distinct transportation modes are presented, illustrating the nuanced evaluation of each mode's performance within the multimodal transportation system. Each transportation mode is represented by a stacked bar showing the division of MDP rewards for three different actions. The rewards are derived from the integration of DEA efficiency scores, signifying the overall effectiveness of each mode based on predefined objectives. The normalization of rewards ensures a consistent and comparable scale, allowing for a meaningful analysis of the relative performance of different transportation modes. This figure provides a visual representation of how the multimodal transportation system is assessed holistically, considering factors such as infrastructure cost, fuel consumption, and employee-related metrics.

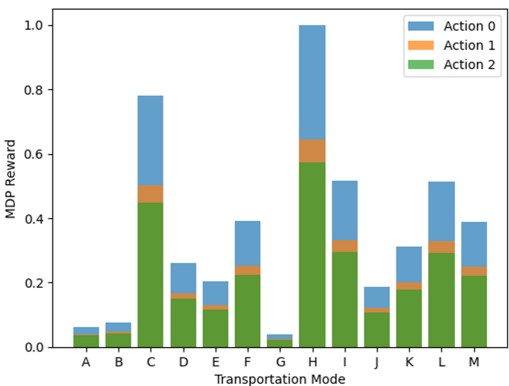

**Figure 3.** MDP reward for different actions taken for 13 transportation modes.

Figure 4 offers valuable insights into the decision-making process by showcasing the actions taken against the optimal transportation policy for each DMU. The optimal policy is determined through the combination of DEA efficiency scores and MDP rewards, resulting in a strategic approach to mode selection for each DMU. This figure provides a clear visualization of how the proposed framework guides decision-makers in choosing transportation modes that balance efficiency, cost-effectiveness, and environmental considerations. It serves as a practical tool for stakeholders to understand the recommended actions based on the integrated evaluation, facilitating informed and data-driven decision-making in the complex landscape of multimodal transportation.

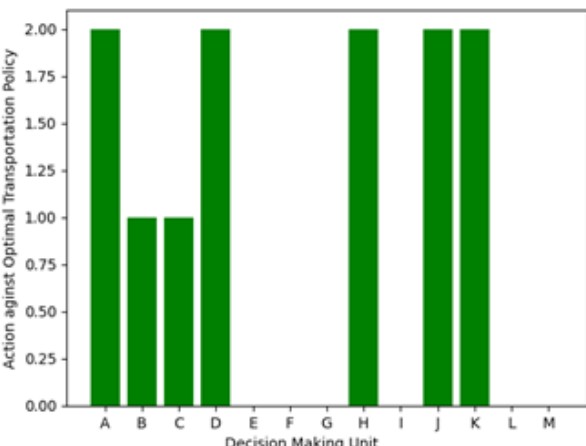

**Figure 4.** Actions against optimal transportation policy for each DMU.

The actions represented by 0, 1, and 2 in Figure 4 correspond to different strategies regarding the selection of transportation modes for each DMU. Action 0: Maintain Status Quo/No Change signifies that, according to optimal policy, the decision is to continue with the existing transportation modes and strategies without making any significant alterations. It suggests that the current combination of transportation modes for the DMU is considered optimal based on the integrated evaluation of DEA efficiency scores and MDP rewards. In view of the foregoing, Action 1: Explore New Transportation Modes reflects a more radical decision by exploring new transportation scenarios. This action implies a willingness to experiment with alternative modes of transportation, aiming to discover more efficient, cost-effective, or sustainable options. It suggests a forward-looking and innovative stance toward enhancing the overall transportation strategy. Moreover, Action 2: Adjust Transportation Strategy represents a decision to modify the contemporary transportation strategy for the DMU. This adjustment could involve changes in the mix of transportation modes, possibly emphasizing more efficient, cost-effective, or environmentally friendly modes based on dynamic evaluation. It indicates a proactive approach to optimize the transportation strategy in response to evolving conditions.

Figure 5 presents a comparative analysis among the proposed methods, Multicast Routing with Multiple QoS Parameters (MRMQoS), and the Minimum Cost Steiner Tree Model (MCSTM) concerning cost, delay, profit, computational time, and $CO_2$ emissions needed for multicast tree construction. The MCSTM model establishes a minimum cost Steiner tree without considering the total delay and profit associated with the resulting tree. It is noteworthy that the proposed method exhibits a total cost approximately 29% higher than that of MRMQoS, and the delay is nearly 26% less than that of MCSTM. However, MRMQoS shows an 8.3% higher profit compared to our proposed method, while our proposed scheme exhibits an 11.7% higher profit compared to MCSTM. In terms of computational time, our method achieves an average CPU time, positioned between MCSTM and MRMQoS. Specifically, the CPU time of MCSTM is about 1.6% better than our proposed scheme, and our proposed scheme shows a 9.5% improved computational time in comparison to MRMQoS (given that our approach is based on determining the

relative efficiency of arcs before solving the integer model). It is pertinent to mention that our proposed method generates reasonably cost-effective multicast trees while exhibiting average delay and profit, thus contributing to an overall superior performance compared to MRMQoS and MCSTM.

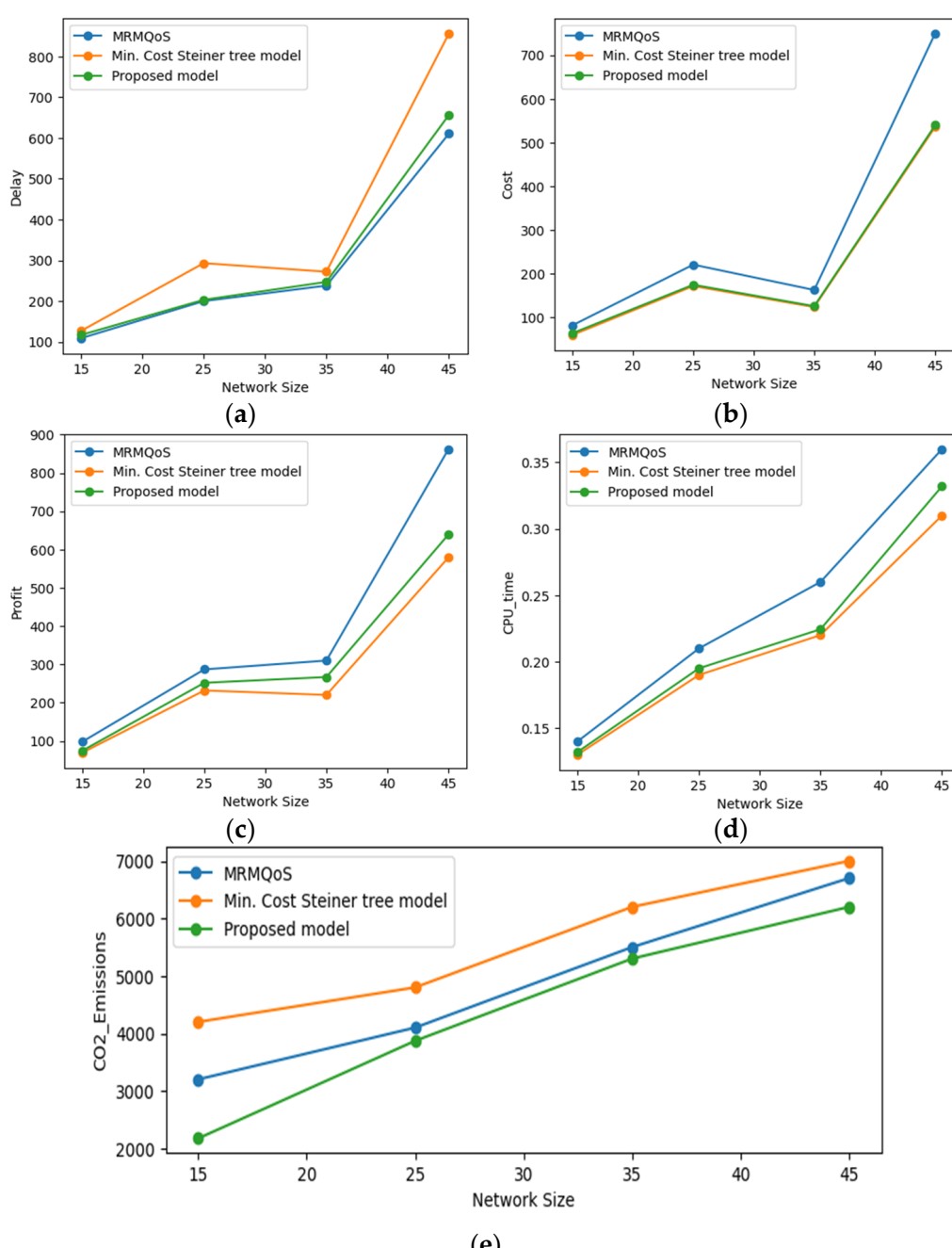

**Figure 5.** Comparison of proposed model, MRMQoS, and MCSTM in terms of cost, delay, profit, and CPU time against network size. (**a**) Cost, (**b**) Delay, (**c**) Profit, (**d**) CPU time, and (**e**) $CO_2$ emissions.

In terms of $CO_2$ emissions, the proposed model consistently outperforms the other models. For instance, in a network of size 15, the proposed model demonstrated significantly lower $CO_2$ emissions (2173 units) compared to MRMQoS (3200 units) and MCSTM (4200 units). This trend persisted across various network sizes, with the proposed model consistently exhibiting reduced $CO_2$ emissions in comparison to both MRMQoS and MCSTM. The percentage decrease in $CO_2$ emissions achieved by the proposed model is 7.26%

and 31.25% when compared against MRMQoS and MCSTM for a network size of 25. These percentages highlight the environmental benefits of the proposed model, showcasing its capacity to contribute to a more sustainable and eco-friendly transportation system. Overall, the proposed model's superior performance in minimizing $CO_2$ emissions underscores its efficacy and potential for promoting environmentally conscious decision-making in multimodal transportation networks. The performance improvement gain of the proposed model against MRMQoS and MCSTM is given in Table 2.

**Table 2.** Performance improvement gain of proposed model against MRMQoS and MCSTM.

| Network Size | MRMQoS | | | | | MCSTM | | | | |
|---|---|---|---|---|---|---|---|---|---|---|
| | Cost | Delay | Profit | CPU Time | $CO_2$ Emissions | Cost | Delay | Profit | CPU Time | $CO_2$ Emissions |
| 15 | −23.53% | −5.8% | −10.5% | 14.3% | 32.1% | 0.2% | 0.3% | 0.2% | 0% | 48.5% |
| 25 | −19.04% | 0.1% | −8.3% | 9.5% | 7.26% | 0% | 32.9% | 11.7% | −1.6% | 31.25% |
| 35 | −29.03% | −2.6% | −8.2% | 15.4% | 4.66% | 0% | 13.2% | 20.8% | −1.3% | 18.3% |
| 45 | −31.37% | −5.9% | −28.6% | 11.1% | 6.15% | 0.1% | 30.5% | 11% | −10% | 13.9% |

In certain instances, the imposition of QoS constraints on a chosen multicast tree often results in its rejection. Consequently, in the subsequent experiments, we explored an extension of our proposed approach wherein the overall weight of a multicast tree must also conform to a budget constraint. This problem is commonly identified as a constrained Steiner tree problem, well-known for its NP-hard nature [31]. A specific manifestation of this model is widely acknowledged as a constrained shortest path problem, a crucial challenge in optimization domains such as transportation, crew scheduling, network routing, and communication networks [32]. The constrained shortest path problem entails identifying the shortest path that adheres to specific constraints, such as delay and cost. Our proposed method was initially employed to calculate the cost, delay, and profit of the selected multicast tree. Subsequently, these metrics were compared with the multicast tree resulting from the variation in MRMQoS using two distinct strategies, with the aim of evaluating the solution quality of the proposed method. Figure 6 visually illustrates the cost comparison of multicast trees obtained for the proposed method and the variable MRMQoS, along with their two strategies. The results align with our expectations, highlighting the proposed method's capacity to discover optimal multicast trees within a reasonable timeframe.

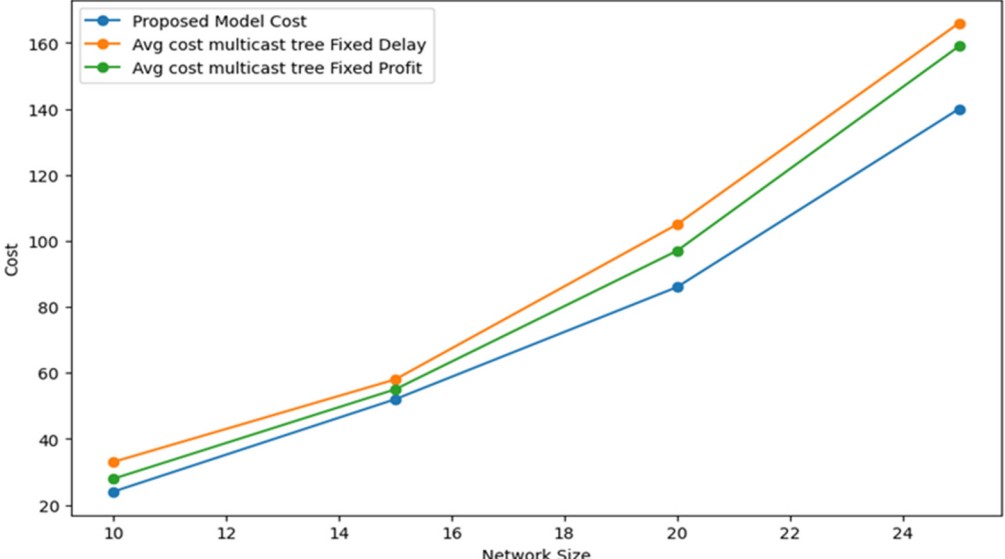

**Figure 6.** Comparison of proposed model with avg. cost multicast tree fixed delay and avg. cost multicast tree fixed profit.

In the initial approach, we evaluated the mean cost of the multicast tree resulting from variations in MRMQoS, where the delay parameter D was held constant, while the profit parameter P was adjusted within the range of $P_{DEA} - 10$ to $P_{DEA} + 10$. The solution quality is notably influenced by the profit requirement P, and surpassing a specific threshold renders the acquisition of the primal solution unfeasible. Subsequently, in the second approach, we maintained the profit parameter P at the $P_{DEA}$ level and varied the delay parameter D from $D_{DEA} - 10$ to $D_{DEA} + 10$. The proposed method consistently outperforms both strategies employed in MRMQoS, specifically the multicast tree with fixed delay and fixed profit, showcasing performance enhancements with reductions of up to 18% and 15%, respectively. Crucially, MRMQoS encounters challenges in networks exceeding 25 nodes due to prolonged running times, whereas our proposed method efficiently identifies optimal solutions within a reasonable timeframe, thus surpassing the performance of both MRMQoS strategies. The performance improvement gain of the proposed model against the avg. cost multicast tree fixed delay and fixed profit is given in Table 3.

**Table 3.** Performance improvement gain of proposed model against avg. cost multicast tree fixed delay and fixed profit.

| Network Size | Avg. Cost Multicast Tree with Fixed Delay | Avg. Cost Multicast Tree with Fixed Profit |
|---|---|---|
| 10 | 37.8% | 23.3% |
| 15 | 16.4% | 11.5% |
| 20 | 21.9% | 15.2% |
| 25 | 24.2% | 15.8% |

## 5. Conclusions

Our research introduces a robust decision-making framework for multimodal transportation, integrating DEA and MDP based on HMM. Utilizing raw data from the Ministry of Transportation, Iraq, our iterative algorithm refines solutions in the intricate supply chain landscape. The incorporation of DEA efficiency scores provides a holistic performance assessment for DMUs, influencing mode selection based on infrastructure cost, fuel consumption, employee metrics, and environmental impact. The DEA efficiency scores, MDP rewards, and optimal policy decisions can facilitate stakeholders with actionable intelligence for mode selection. Our comparative analysis showcases the superiority of our proposed model against Multicast Routing with Multiple QoS Parameters (MRMQoS) and the Minimum Cost Steiner Tree Model (MCSTM) across key metrics. While our approach presents a total cost approximately 29% higher than MRMQoS, it boasts a nearly 26% reduction in delay compared to MCSTM. However, MRMQoS shows an 8.3% higher profit than our method, whereas our proposed scheme exhibits an 11.7% higher profit compared to MCSTM. Regarding computational time, our method achieves an average CPU time positioned between MCSTM and MRMQoS, with MCSTM showing about a 1.6% better CPU time than our approach, and our method displaying a 9.5% improvement in computational time compared to MRMQoS. Additionally, in terms of $CO_2$ emissions, the proposed model consistently outperforms the other models across various network sizes. The proposed model demonstrated significantly lower $CO_2$ emissions (2173 units) compared to MRMQoS (3200 units) and MCSTM (4200 units) in a network of size 25. The percentage decrease in $CO_2$ emissions achieved by the proposed model is 7.26% and 31.25% when compared against MRMQoS and MCSTM for a network size of 25, respectively. Our proposed method excels in addressing QoS constraints, demonstrating resilience and efficiency in multicast tree construction, further underscoring its practicality and effectiveness.

## 6. Future Work

Looking ahead, future recommendations include both refining the proposed framework and broadening its applicability. Beyond the immediate context of Iraq's transportation network, the methodology integrating DEA efficiency scores and MDP holds promise

for optimizing multimodal transportation systems in various geographic regions. Customization of input parameters and constraints to match the characteristics of different transportation networks enables the framework's effective application to similar challenges encountered in other countries or regions. This adaptability underscores the potential for widespread implementation and impact, offering a versatile solution to enhance transportation efficiency, cost-effectiveness, and environmental sustainability globally.

Moreover, the insights gained from analyzing the performance of different transportation modes and optimizing strategies in Iraq's context have broader implications for decision-making processes in diverse geographic contexts. Understanding the trade-offs between transportation costs, delay costs, and environmental impacts provides valuable guidance for policymakers and stakeholders worldwide. By extrapolating these insights, transportation authorities can make informed decisions regarding infrastructure development, mode selection, and resource allocation to address common challenges faced by transportation networks globally.

To facilitate the adaptation and implementation of the optimization framework in various regions, future research efforts should focus on refining the model and exploring potential extensions to address additional factors or constraints relevant to transportation optimization. Collaboration with local transportation authorities, research institutions, and industry stakeholders in different geographic areas will be instrumental in tailoring the framework to address specific transportation challenges and priorities. Additionally, sharing methodologies, findings, and best practices through academic publications, conferences, and collaborative research projects will contribute to global knowledge exchange and capacity building in transportation optimization, fostering innovation and progress in the field.

**Author Contributions:** Conceptualization, M.R.A. and K.B.; Data curation, M.R.A., K.B., S.G. and A.J.O.; Formal analysis, M.R.A., K.B. and S.G.; Investigation, M.R.A. and K.B.; Methodology, M.R.A. and K.B.; Project administration, M.R.A., K.B. and S.G.; Supervision, M.R.A. and K.B.; Visualization, M.R.A., K.B. and A.J.O.; Writing—original draft, M.R.A. and K.B.; Writing—review & editing, M.R.A., K.B., S.G. and A.J.O. All authors have read and agreed to the published version of the manuscript.

**Funding:** This research received no external funding.

**Data Availability Statement:** Data were collected from four places in the Iraqi Ministry of Transport, which is the General Company for (land, train, water, and air transport) [30].

**Conflicts of Interest:** The authors declare no conflicts of interest.

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
