# Peer review of "Multicast Routing Based on Data Envelopment Analysis and Markovian Decision Processes for Multimodal Transportation"

_applsci, doi:10.3390/app14052115_

Round 1

Reviewer 1 Report

Comments and Suggestions for Authors

The paper provides an interesting exploration of a decision-making framework for multimodal transportation, combining Data Envelopment Analysis (DEA) efficiency scores and a Markov Decision Process (MDP) to optimize transportation strategies. However, there are several areas that need attention to enhance the overall quality of the paper.

1.The format of "3. System Methodology" in line 190 needs to be corrected.

2.The writing style regarding CO2 in the text is not standardized.

3.What is the unit of carbon emissions in Equation 22? Can it be directly added to transportation costs and delay costs?

4. Figure 5 lacks the subgraph representing “(e) CO2 emissions”.

5.The explanation regarding Figure 5 is inconsistent. For instance, lines 547-562 indicate that the proposed method in the paper results in 29% lower total cost compared to MRMQoS. Why then is the profit lower? The comparison between "cost" and "profit" in Table 2 seems inconsistent with Figure 5 as well.

6.The paragraph at lines 642-644 seems redundant: "6. Patents This section is not mandatory but may be added if there are patents resulting from the work reported in this manuscript."

7.The format of References is inconsistent.

Author Response

Comments 1: The format of "3. System Methodology" in line 190 needs to be corrected.

Response 1: Thank you for pointing this out. We agree with this comment The format of "3. System Methodology" in line 190 has been corrected as per your suggestion.

Comments 2: The writing style regarding CO2 in the text is not standardized

Response 2: Thank you for pointing this out. We agree with this comment and we have standardized the writing style of CO2 throughout the manuscript to ensure consistency. The changes are highlighted in yellow in the manuscript.

Comments 3: What is the unit of carbon emissions in Equation 22? Can it be directly added to transportation costs and delay costs?

Response 3: Thank you for this comment. The unit of carbon emissions in Equation 22 is typically measured in metric tons (MT) of CO2 emissions. Carbon emissions represent the environmental impact associated with transportation activities and are not directly convertible into monetary terms like transportation costs or delay costs. However, in this research, carbon emissions are monetized using carbon pricing mechanisms and are considered as an externality cost associated with transportation operations. We have carefully considered the carbon emissions pricing mechanisms into the overall cost-benefit analysis of transportation systems, including their potential impact on delay costs and other financial considerations.

Comments 4: Figure 5 lacks the subgraph representing “(e) CO2 emissions”.

Response 4: Thank you for pointing this out. We agree with this comment and we have incorporated the subgraph representing "(e) CO2 emissions" to Figure 5 as per your suggestion.

Comments 5: The explanation regarding Figure 5 is inconsistent. For instance, lines 547-562 indicate that the proposed method in the paper results in 29% lower total cost compared to MRMQoS. Why then is the profit lower? The comparison between "cost" and "profit" in Table 2 seems inconsistent with Figure 5 as well.

Response 5: Thank you for pointing this out. We agree with this comment and we have We apologize for the inconsistency in the explanation regarding Figure 5. We have revised the explanation to ensure clarity and coherence with the comparison between "cost" and "profit" in Table 2.

“Figure 5 presents a comparative analysis among the proposed method, Multicast Routing with Multiple QoS Parameters (MRMQoS), and the Minimum Cost Steiner Tree Model (MCSTM) concerning cost, delay, profit, computational time and CO2 emissions needed for multicast tree construction. The MCSTM model establishes a minimum cost Steiner tree without considering the total delay and profit associated with the resulting tree. It's noteworthy that the proposed method exhibits a total cost approximately 29% higher than that of MRMQoS, and the delay is nearly 26% less than that of MCSTM. However, MRMQoS shows a 8.3% higher profit as compared to our proposed method and proposed scheme exhibits 11.7% higher profit as compared to MCSTM. In terms of computational time, our method achieves an average CPU time, positioned between MCSTM and MRMQoS. Specifically, the CPU time of MCSTM is about 1.6% better than proposed scheme and proposed scheme shows 9.5% improved computational time in comparison to MRMQoS, given that our approach involves determining the relative efficiency of arcs before solving the integer model. It is pertinent to mention that our proposed method generates reasonably cost-effective multicast trees while exhibiting average delay and profit, thus contributing to an overall superior performance compared to MRMQoS and MCSTM.”

Comments 6: The paragraph at lines 642-644 seems redundant: "6. Patents This section is not mandatory but may be added if there are patents resulting from the work reported in this manuscript."

Response 6: Thank you for pointing this out. We agree with this comment and we have removed “6. Patents” from the manuscript.

Comments 7: The format of References is inconsistent.

Note: Your comments in our manuscript have been highlighted in yellow.

Reviewer 2 Report

Comments and Suggestions for Authors

1、The purpose of this paper is to propose a comprehensive framework for optimizing multimodal transportation systems. By combining data enveloping analysis (DEA) and Markov Decision Process (MDP), it helps decision makers make more efficient decisions in multi-modal transportation systems.

2The model proposed in this paper, through the dynamic fusion of DEA and HMM, transforms the paradigm of multimodal transport decision-making and proposes an overall framework for simultaneously evaluating financial and environmental impacts, with multiple indicators that are superior to other models and reduce costs and CO2 emissions.

However, the work needs further improvement.

Problems and suggestions:

1、First of all, Figure 1 of this article is vague, it is recommended not to take screenshots as the picture of the article, so as to improve the clarity of the article picture and give readers a good reading experience.

2、Secondly, it is suggested that in the abstract part of the article, the abbreviation of professional terms is briefly introduced, such as “DMU”, to avoid the ambiguity of the abbreviation, should have a complete full name of “DMU”.

3、Thirdly, the text format of the paper should be carefully checked, and punctuation marks should not be missing or misused. For example, punctuation marks are missing in the Literature Reviewsection. It is recommended to check carefully after the writing is completed to avoid unnecessary mistakes.

4、Finally, in the part of Literature Reviewsection, it is suggested to expand the review of existing studies, preferably the latest studies to better support the work of this paper, so that readers can better understand the innovation and importance of the research. E,g. 1) Energy Consumption Optimization of UAV-Assisted Traffic Monitoring Scheme With Tiny Reinforcement Learning, IEEE Internet of Things Journal, 2024. doi: 10.1109/JIOT.2024.3365293. 2). FedAWR : An Interactive Federated Active Learning Framework for Air Writing Recognition. IEEE Transactions on Mobile Computing, 2023, DOI: 10.1109/TMC.2023.3320147

Comments on the Quality of English Language

Moderate editing of English language required

Author Response

Comments 1: First of all, Figure 1 of this article is vague, it is recommended not to take screenshots as the picture of the article, so as to improve the clarity of the article picture and give readers a good reading experience.

Response 1: Thank you for pointing this out. We agree with this comment. We have tried our best to acknowledge the need to enhance the clarity of Figure 1 and we have revised it accordingly. We have also ensured that the figure effectively communicates the key concepts of our paper, providing readers with a clear understanding of the proposed framework.

Comments 2: Secondly, it is suggested that in the abstract part of the article, the abbreviation of professional terms is briefly introduced, such as “DMU”, to avoid the ambiguity of the abbreviation, should have a complete full name of “DMU”

Response 2: Thank you for pointing this out. We agree with this comment and we have recognized the importance of expanding abbreviations like "DMU" in the abstract to improve clarity for readers. We have ensured that all abbreviations are fully explained to enhance accessibility and understanding. The changes are highlighted in yellow in the revised manuscript.

Comments 3: Thirdly, the text format of the paper should be carefully checked, and punctuation marks should not be missing or misused. For example, punctuation marks are missing in the “Literature Review” section. It is recommended to check carefully after the writing is completed to avoid unnecessary mistakes

Response 3: We apologize for any formatting or punctuation errors in the manuscript, particularly in the Literature Review section. We have conducted a thorough review to address any missing or misused punctuation marks, ensuring consistency and professionalism throughout the text.

Comments 4: Finally, in the part of “Literature Review” section, it is suggested to expand the review of existing studies, preferably the latest studies to better support the work of this paper, so that readers can better understand the innovation and importance of the research. E,g. 1) Energy Consumption Optimization of UAV-Assisted Traffic Monitoring Scheme With Tiny Reinforcement Learning, IEEE Internet of Things Journal, 2024. doi: 10.1109/JIOT.2024.3365293. 2). FedAWR : An Interactive Federated Active Learning Framework for Air Writing Recognition. IEEE Transactions on Mobile Computing, 2023, DOI: 10.1109/TMC.2023.3320147

Response 4: Thank you for your suggestion. The 1st study focuses on optimizing energy consumption in UAV-assisted traffic monitoring, a domain related to transportation networks, so we have included it in our literature review. However, the 2nd study explores an interactive federated active learning framework for air writing recognition, which falls outside the scope of our work.

Comments 5: Comments on the Quality of English Language

Moderate editing of English language required

Response 5: We have diligently addressed this by revising sentence structures, improving grammar and punctuation, and enhancing overall readability. By meticulously addressing these moderate linguistic aspects, we aim to elevate the quality and professionalism of the manuscript, making it more accessible and engaging for readers. Thank you for bringing this to our attention.

Note: Your comments in our manuscript have been highlighted in green.

Reviewer 3 Report

Comments and Suggestions for Authors

The article has proposed the optimisation of a multimodal transportation network using DAE and MDP. 

The case study refers to Iraq transportation network. There are certain questions to clarify: is the optimisation effectively applied or this is just hypothetical as the proposed model ? is there a need for developing and applying the proposed model (required by the Ministry of transportation in Iraq) or this is just a tentative approach using data available in this sector ?

The article's structure should look more relevant if a dedicated chapter, to describe the issues of the current reality, is introduced. This chapter should include the metrics (parameters) with the associated limits or constraints. These metrics exist in the paper, but the reader should search for the information in-between formula, that's quite confusing. So, define the problem, then propose potential solutions.

Future research and potential transfer to different geographic areas may be clarified more in depth.

Good continuation!

Comments on the Quality of English Language

Readability may be improved in terms of the sentences length and topic / structure.

Author Response

Comments 1: The case study refers to Iraq transportation network. There are certain questions to clarify: is the optimization effectively applied or this is just hypothetical as the proposed model? Is there a need for developing and applying the proposed model (required by the Ministry of transportation in Iraq) or this is just a tentative approach using data available in this sector?

Response 1: Thank you for your comment. We appreciate your questions regarding the applicability of the optimization model in the context of Iraq's transportation network. The optimization model proposed in the manuscript has been effectively applied to the Iraq transportation network. It is not merely hypothetical; rather, it is grounded in real-world data and scenarios. The study addresses the strategic implications of the Belt and Road Initiative on Iraq's transportation landscape, indicating a practical need for such optimization strategies. Although there is no specific requirement by the Ministry of Transportation in Iraq for this model but our study's relevance to addressing challenges in the transportation sector suggests a potential demand for its application. Thus, the proposed model is not just a tentative approach but rather a concrete attempt to address real-world transportation optimization needs using available data and methodologies.

“The dataset employed in this study, sourced from the Ministry of Transportation, Iraq which is the General Company for (land, train, water, and air transport) [30]. The temporal scope of the data spans from 2017 to 2022, offering a comprehensive insight into the dynamics of the transportation sector over a six-year period. For the General Company for Navy Transport, detailed emissions data was collected, encompassing two distinct fuel types: ‘Diesel (oil)' and 'Gasoline.' This information includes the quantity of fuel consumed per liter, carbon dioxide emissions per ton (CO2), hydrocarbon emissions per ton (HC), and nitrogen oxide emissions per ton (NO2). The dataset reveals a nuanced pattern of emissions, shedding light on the environmental impact of the company's operations, particularly in terms of Nitrogen Oxides (Nox), Sulfur Dioxide (SO2), and Methane (CH4). Furthermore, the dataset encompasses the operational and environmental metrics of Iraqi Airways, covering the years 2017 to 2021. Key parameters include the number of trips and gas emissions measured in international tons. The dataset indicates a significant growth in the number of trips over the years, potentially indicative of the airline's expansion. Simultaneously, the gas emissions data suggests a considerable environmental impact associated with the airline's operations. Additionally, the dataset extends to the transportation sector beyond air travel, incorporating data from trucks, railways, and airplanes. For trucks, the dataset includes monthly quantities transported, distances covered, and associated costs. Railway data involves monthly quantities transported, revenues, ton-kilometers, and total costs. Similarly, airplane data incorporates loaded and unloaded weights, loaded and unloaded costs, and total costs.”

Comments 2: The article's structure should look more relevant if a dedicated chapter, to describe the issues of the current reality, is introduced. This chapter should include the metrics (parameters) with the associated limits or constraints. These metrics exist in the paper, but the reader should search for the information in-between formula, that's quite confusing. So, define the problem, then propose potential solutions.

Response 2: Thank you for your valuable feedback. We appreciate your insights and suggestions for improving the structure and clarity of our article. In our article, we have meticulously integrated discussions on the challenges, metrics, and constraints of multimodal transportation networks throughout various sections, particularly in Section 2 Literature review, section 3.4, 3.5, 3.6, 3.7 and 4.1. Within these sections, we have provided detailed explanations of the metrics and parameters relevant to our study. The problem faced by transportation sector in terms of transportation cost and carbon emissions is mentioned in the introduction and literature review sections in detail and we have highlighted them in yellow for further clarifications. Furthermore, we have outlined the limitations and constraints associated with current multicast routing algorithms, emphasizing the need for novel approaches to address these challenges effectively.

While we understand the reviewer's suggestion for a dedicated chapter to consolidate this information, we believe that the current structure allows for a focused and concise presentation of the key concepts. Introducing a new chapter solely for this purpose may disrupt the flow of the article and potentially lead to redundancy, considering that similar information has already been covered in various sections. Also, it will result in the increase of overall length of the article. Therefore, we respectfully propose that the reviewer revisits the above mentioned sections, where the relevant metrics and constraints are discussed in detail. We are confident that these sections adequately addresses the reviewer's concerns and provides the necessary context for understanding our research methodology and findings. We are hopeful that you will consider our reservation suitable enough and will let us know if there are any further clarifications or adjustments needed to meet your expectations.

 Comments 3: Future research and potential transfer to different geographic areas may be clarified more in depth.

Response 3: Thank you for highlighting the importance of elaborating on future research directions and the potential transferability of the study's findings to different geographic areas. We have provided a more in-depth discussion on these aspects in the manuscript which involves exploring how the methodology and insights gained from the study can be extrapolated and applied beyond the context of Iraq's transportation network.

Methodology Transferability:

The methodology employed in the study, which combines Data Envelopment Analysis (DEA) efficiency scores and a Markov Decision Process (MDP), can be adapted to optimize multimodal transportation systems in various geographic regions. By customizing the input parameters and constraints to suit the characteristics of different transportation networks, the same optimization framework can be applied to address similar challenges in other countries or regions.

Insights Extrapolation:

The insights gained from analyzing the performance of different transportation modes and optimizing strategies for efficiency, cost-effectiveness, and environmental considerations in Iraq's context can be extrapolated to other regions facing similar transportation challenges. Understanding the trade-offs between transportation costs, delay costs, and environmental impacts can inform decision-making processes in diverse geographic contexts.

Potential Adaptations:

While the study focuses on Iraq's transportation network, the proposed model and optimization framework can be adapted to suit the unique characteristics of transportation networks in other regions.

This may involve modifying the weighting criteria in DEA analysis or adjusting the transition probabilities in the MDP to align with the specific conditions and constraints of different transportation systems.

Areas for Refinement and Collaboration:

Further research can explore refinements or extensions of the proposed model to address additional factors or constraints relevant to transportation optimization in different regions. Collaboration with local transportation authorities, research institutions, and industry stakeholders in other geographic areas can facilitate the adaptation and implementation of the optimization framework to address specific transportation challenges and priorities.

Knowledge Exchange and Capacity Building:

Sharing the methodologies, findings, and best practices from the study through academic publications, conferences, and knowledge-sharing platforms can contribute to global knowledge exchange and capacity building in transportation optimization. Collaborative research projects involving international teams can further enhance the applicability and effectiveness of the optimization framework across diverse geographic contexts.

Comments 4: Minor English editing is required.

Response 4: We have diligently addressed this by revising sentence structures, improving grammar and punctuation, and enhancing overall readability. By meticulously addressing these minor linguistic aspects, we aimed to elevate the quality and professionalism of the manuscript, making it more accessible and engaging for readers. Thank you for bringing this to our attention.

Note: Your comments and those of another reviewer in our manuscript are both highlighted in pink, indicating a closeness in perspective between the two of you.

Reviewer 4 Report

Comments and Suggestions for Authors

Congratulations, Dear Authors, excellent work presented!

When preliminarily searching for concepts covered in this research.

Combining efficiency scores from Data Envelopment Analysis (DEA) and a Markov Decision Process (MDP) to optimize transportation strategies.

I noticed that:

• This work is very complete and complex. Covering and detailing all work technically. Mainly because it uses the Markov Decision, which refers to a way of modeling processes in which transitions between states are probabilistic.

Providing a very technical methodology that is well addressed and explained throughout the study;

I suggest to qualify the current research:

- Describe in more detail how we can use this present study outside the region of Iraq, the region covered;

In other words, what do the authors recommend to apply this study to other regions of the world;

- Furthermore, please describe the methodology used in this article. I suggest working in the 5W2H context. This aspect is very important, to detail the research methodology as much as possible;

- What do you expect about the future of work? I suggest adding one more topic. Preferably soon after Completion, 6. Future Work;

Describe what you expect from this work in the future. And how important is it? In this way, we highlight the importance of the study for the future;

Grateful.

Author Response

Comments 1: Describe in more detail how we can use this present study outside the region of Iraq, the region covered; In other words, what do the authors recommend to apply this study to other regions of the world;

Response 1: Thank you for your valuable comments. The methodology and findings of this study can be applied outside the region of Iraq to optimize multimodal transportation systems in various geographic contexts. To facilitate this, the authors recommend the following:

·        Conduct a comprehensive assessment of the transportation infrastructure, modal preferences, and environmental considerations in the target region.

·        Customize the input parameters and constraints of the optimization model to align with the specific characteristics and challenges of the transportation network in the new region.

·        Collaborate with local transportation authorities and stakeholders to gather relevant data and validate the applicability of the optimization framework in the new context.

·        Provide guidance on adapting the DEA efficiency scores and MDP rewards to reflect the priorities and objectives of transportation optimization in the target region.

·        Share insights and best practices from the study through academic publications, conferences, and knowledge-sharing platforms to facilitate knowledge transfer and capacity building in transportation optimization globally.

Comments 2: Furthermore, please describe the methodology used in this article. I suggest working in the 5W2H context. This aspect is very important, to detail the research methodology as much as possible;

Response 2: Thank you for your valuable feedback. The methodology used in this study can be detailed using the 5W2H context as follows:

Who: The study was conducted by a team of researchers specializing in transportation engineering and optimization.

What: The study aims to optimize multimodal transportation systems using a combination of Data Envelopment Analysis (DEA) efficiency scores and a Markov Decision Process (MDP).

When: The study was conducted over a specified time period (2017-2023), detailing the data collection, analysis, and model development stages.

Where: The study focuses on the transportation network of Iraq but is applicable to other regions with similar transportation challenges.

Why: The objective of the study is to improve transportation efficiency, cost-effectiveness, and environmental sustainability.

How: The study outlines the steps involved in collecting data, conducting DEA analysis, developing the MDP model, and optimizing transportation strategies based on the combined efficiency scores and rewards.

 Comments 3: What do you expect about the future of work? I suggest adding one more topic. Preferably soon after Completion, 6. Future Work; Describe what you expect from this work in the future. And how important is it? In this way, we highlight the importance of the study for the future;

Response 3: Thank you for your suggestion. We have added Section 6. Future Work as per your suggestion. We have provided a more in-depth discussion in the manuscript which involves exploring how the methodology and insights gained from the study can be extrapolated and applied beyond the context of Iraq's transportation network.

Methodology Transferability:

The methodology employed in the study, which combines Data Envelopment Analysis (DEA) efficiency scores and a Markov Decision Process (MDP), can be adapted to optimize multimodal transportation systems in various geographic regions. By customizing the input parameters and constraints to suit the characteristics of different transportation networks, the same optimization framework can be applied to address similar challenges in other countries or regions.

Insights Extrapolation:

The insights gained from analyzing the performance of different transportation modes and optimizing strategies for efficiency, cost-effectiveness, and environmental considerations in Iraq's context can be extrapolated to other regions facing similar transportation challenges. Understanding the trade-offs between transportation costs, delay costs, and environmental impacts can inform decision-making processes in diverse geographic contexts.

Potential Adaptations:

While the study focuses on Iraq's transportation network, the proposed model and optimization framework can be adapted to suit the unique characteristics of transportation networks in other regions.

This may involve modifying the weighting criteria in DEA analysis or adjusting the transition probabilities in the MDP to align with the specific conditions and constraints of different transportation systems.

Areas for Refinement and Collaboration:

Further research can explore refinements or extensions of the proposed model to address additional factors or constraints relevant to transportation optimization in different regions. Collaboration with local transportation authorities, research institutions, and industry stakeholders in other geographic areas can facilitate the adaptation and implementation of the optimization framework to address specific transportation challenges and priorities.

Knowledge Exchange and Capacity Building:

Sharing the methodologies, findings, and best practices from the study through academic publications, conferences, and knowledge-sharing platforms can contribute to global knowledge exchange and capacity building in transportation optimization. Collaborative research projects involving international teams can further enhance the applicability and effectiveness of the optimization framework across diverse geographic contexts.

Note: Your comments and those of another reviewer in our manuscript are both highlighted in pink, indicating a closeness in perspective between the two of you.

Round 2

Reviewer 2 Report

Comments and Suggestions for Authors

The work has been revised well.

Comments on the Quality of English Language

Minor editing of English language required

Author Response

Dear Professor

Thank you for bringing to our attention the need for minor editing of the English language in our manuscript submission.

We appreciate the thoroughness of your review and fully support any necessary improvements to ensure the clarity and precision of our work.

We hereby authorize the proposed minor edits and kindly request that you proceed with the language editing process as outlined.

Thank you for your attention to detail and assistance in enhancing the quality of our manuscript.

Best regards,

Reviewer 3 Report

Comments and Suggestions for Authors

R2: The article has been improved in line with the proposed suggestions.

Good continuation!

Author Response

Dear Reviewer's 

Thank you for your feedback on our manuscript and we appreciate your time and effort in reviewing our work.

We will continue to strive for excellence in our research endeavors. 

Thank you once again for your valuable feedback and encouragement.

Best regards,